# SAPE: Spatially-Adaptive Progressive Encoding for Neural Optimization

**Amir Hertz**
Tel Aviv University
amirhertz@mail.tau.ac.il

**Or Perel**
Tel Aviv University
orr.perel@gmail.com

**Raja Giryes**
Tel Aviv University
raja@tauex.tau.ac.il

**Olga Sorkine-Hornung**
ETH Zurich, Switzerland
sorkine@inf.ethz.ch

**Daniel Cohen-Or**
Tel Aviv University
cohenor@gmail.com

## Abstract

Multilayer-perceptrons (MLP) are known to struggle with learning functions of high-frequencies, and in particular cases with wide frequency bands. We present a spatially adaptive progressive encoding (SAPE) scheme for input signals of MLP networks, which enables them to better fit a wide range of frequencies without sacrificing training stability or requiring any domain specific preprocessing. SAPE gradually unmasks signal components with increasing frequencies as a function of time and space. The progressive exposure of frequencies is monitored by a feedback loop throughout the neural optimization process, allowing changes to propagate at different rates among local spatial portions of the signal space. We demonstrate the advantage of SAPE on a variety of domains and applications, including regression of low dimensional signals and images, representation learning of occupancy networks, and a geometric task of mesh transfer between 3D shapes.

## 1 Introduction

*Neural implicit functions* have recently emerged as a powerful representation paradigm for modeling complex signals. Their continuous nature sets them apart from typical discrete representations (i.e. pixels, voxels, meshes), allowing to capture high resolution details in various domains such as images [41, 43], 3d shapes [31, 6] and radiance fields [30, 32], while retaining a reasonably compact representation. In this formulation, a deep neural network is trained with the goal of faithfully mapping *input coordinates* to a corresponding target domain, effectively learning the representation of signal properties such as magnitude, color, or shape occupancy.

Implementing implicit neural representations with common neural structures, e.g., multilayer pereceptrons with ReLU activations (ReLU MLPs), proves to be challenging in the presence of signals with high frequencies. Consequently, recent works have demonstrated that deep implicit networks benefit from mapping the input coordinates [30, 43], or the intermediate features [41] to positional encodings. That is, before feeding them into a neural layer, they are first transformed to an overparameterized, high dimensional space, typically by multiple periodic functions.

Positional encodings[1] have been shown to enable highly detailed mappings of signals by MLP networks. For example, *Fourier Feature Networks* [43] suggested to map input coordinates of signals to a high dimensional space using sinusoidal functions. In their work, they show that the frequency

---

[1]In this paper, we use the term "positional encodings" in lower case letters to denote the *family* of encoding methods that map coordinates to a higher dimensional space. Not to be confused with the term "Positional Encoding" coined by [30, 43], which refers to a particular mapping scheme in this family.

35th Conference on Neural Information Processing Systems (NeurIPS 2021).

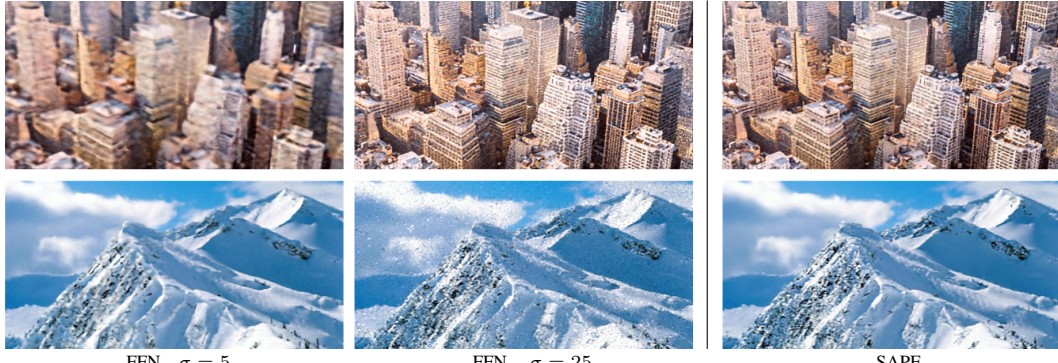

|  FFN $\quad\sigma = 5$  |  FFN $\quad\sigma = 25$  |  SAPE  |

Figure 1: Left: Fourier Features Network (FFN) encoding, tuned to a bandwidth of low frequencies in order to fit the smooth snow areas. The same frequency bandwidth yields blurry buildings in the top image. Middle: FFN tuned to a higher bandwidth to fit to the sharp details of the city. The same bandwidth results in the appearance of noisy artifacts in the mountain image. Right: SAPE is able to fit both examples without extra tuning, using the same choice of frequency bandwidth in both cases.

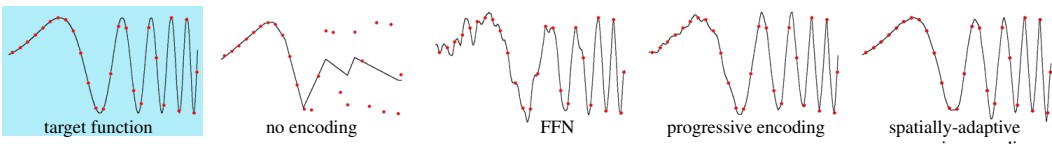

Figure 2: 1D signal regression. **Red**: Samples of positional coordinates as network input, and signal magnitude as labels. **Black**: Predicted implicit signal at inference time. MLPs with "no encoding" struggle to fit high frequency segments (see appendix for train details). Efforts of static positional encoding models (FFN) to fit high frequency areas of the signal introduce noise at the low frequency region. Fitting varying signals with spatial adaptivity allows MLPs to recover all signal frequencies.

of these sinusoidal encodings is the dominant factor in obtaining high quality signal reconstructions. In particular, they present compelling arguments for randomly sampling frequency values from an isotropic Gaussian distribution with a *carefully* selected scale, providing a striking improvement over mapping coordinates directly via standard MLPs.

Despite the success of positional encodings, there are still some concerns left unaddressed: (i) Choosing the right frequency scale requires manual tuning, oftentimes involving a tedious parameter sweep; (ii) The frequency distribution scale may change between different inputs, and accordingly it becomes harder to tune a "one-fits-all" model for signals that are composed of a large range of frequencies (Fig. 1); (iii) Frequencies are selected for the entire input in a global, spatially invariant manner, thus missing an opportunity to better adapt to local high frequencies (Fig. 2).

Our work investigates mitigations to the aforementioned challenges. We study the setting of positional encodings as input to implicit neural networks and present **S**patially-**A**daptive **P**rogressive **E**ncoding (SAPE). SAPE is a policy for learning implicit functions, relying on two core ideas: (i) guiding the neural optimization process by gradually unmasking signal components with increasing frequencies over time, and (ii) allowing the mask progression to propagate at different rates among local spatial portions of the signal space. To govern this process, we introduce a *feedback loop* to control the progression of revealed encoding frequencies as a bi-variate function of time and space.

Our work enables MLP networks to adaptively fit a varying spectrum of fine details that previous methods struggle to capture in a single shot, without involved tuning of parameters or domain specific preprocessing. SAPE excels in learning implicit functions with a large Lipschitz constant, without sacrificing quality of details or optimization stability, in problems that require meticulous configuration to achieve convergence. To highlight the latter, in Section 5.1 we present the tasks of 2D silhouettes deformation and 3D mesh transfer – both require stable optimization from the get-go in order to avoid convergence to sub-optimal local minima.

SAPE is encoding-agnostic: it is not limited to a specific positional encoding type. It can be easily applied to the learning process of coordinate-based neural implicit functions of various domains including images, 2D shapes, 3D occupancy maps and surfaces, showing improvement in all of them.

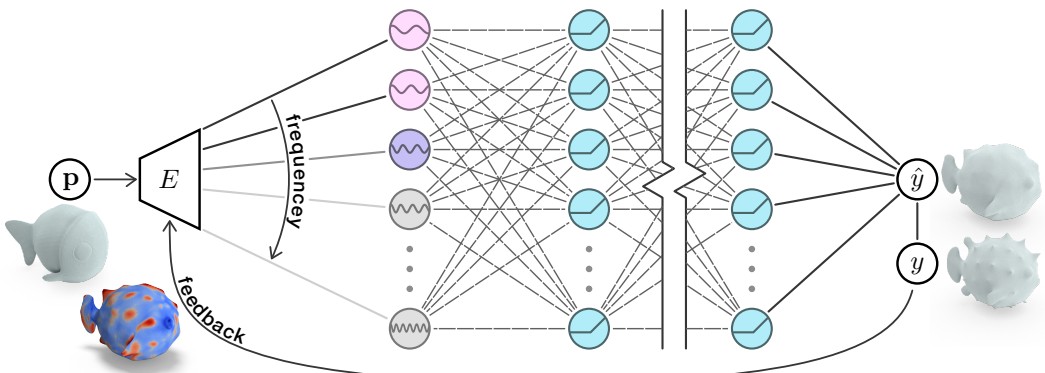

Figure 3: Method overview. Coordinates **p** are fed into an implicit function network, regressing the signal value $\hat{y}$ at position **p** as output. An encoder $E$ maps the input coordinates to a high dimensional embedding space. Our encoding layer then masks the encoded features. In this example, samples **p** are encoded and progressively exposed during the network training, going from low frequency (top node) to high (bottom node). Node colors indicate the various neuron states: on, off, and partially masked. Finally, the loss between output $\hat{y}$ and target $y$ is fed back to encoder $E$, which spatially adapts the encoding mask according to the spatial error, indicated by the heatmap on the pufferfish.

## 2 Related work

**Implicit neural representations.** Recent works show the capability of deep neural networks in representing different functions (e.g., 2D/3D scenes or objects) as an implicit, memory efficient continuous function [8]. These networks are used as a signed distance function (SDF) [1, 15, 31] or as an occupancy network, either binary [6, 28, 33] or soft volumetric density [26, 30, 49].

Several works train such networks to represent a collection of 3D shapes via 3D data supervision [6, 16, 29, 28, 31], reconstruct 3D shapes [5, 11, 13, 14, 18, 42] or infer them from 2D images [22, 37, 40]. Recent works employ spatial data structures to scale the represented shapes size [25, 42].

**Positional encodings (PE).** have been suggested as higher dimensional network input for various purposes. Radial basis function (RBF) networks [7] use weighted sums of embedded RBF encodings due to their symmetric property. [47] used random Fourier Features to map time spans. In natural language processing, Transformers [19, 23, 24, 46] leverage sinusoidal positional encoding to maintain the order of token sequences. Our work differs by focusing on how to encode the inputs to MLPs in order to improve network implicit representations.

The closest works to ours are SIREN [41] and Fourier feature networks (FFN) [43]. SIREN suggests to replace the ReLU activations in the network by periodic ones. FFN [43] encodes the inputs to the network by projecting them to a higher dimensional space using a family of encoding functions. In the appendix we provide an illustration of this approach with some examplary used encodings. The analysis of [43] demonstrates that FFN improves the learning of implicit neural representation compared to other alternatives. A follow-up work in [44] accelerates the training convergence by using a meta-learned initialization of the network weights. The contemporary work of [27] extends SIREN with a modulation network, to allow representing multiple shapes within a single network.

Concurrently to our work, Park et al. [32] extend NERF to non-rigidly deforming scenes, and BARF [21] extends NERF for cases of imperfect camera poses. Both these works show the advantage of employing coarse-to-fine annealing linearly over the frequency bandwidth of the positional encoding. The coarse-to-fine approach bears similarity to our progressive frequency approach. Different from us, these works do not use a feedback loop or spatial encoding, which, as we show in the following, further closes the gap between the regressed function and the ground truth.

As a final remark, SAPE's concept of feedback loop resonates with the idea of "predictive coding" in computational neuroscience [35], suggesting another angle to appreciate the rationale of our method.

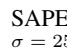
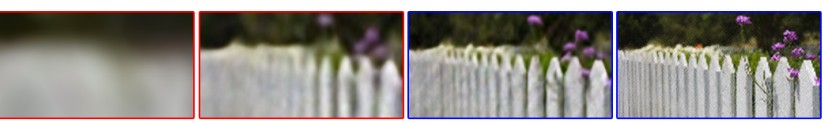

Figure 4: Learning progress of SAPE for implicit representation of a 2D image, showing iterations 75, 150, 300 and 3000 (left to right). The beginning of optimization is dominated by low frequencies: **spectral bias** maintains global optimization stability. As the optimization progresses **high frequency mapping** takes place, and SAPE fits the finer details of the image. See other methods in the appendix.

## 3 Preliminaries

The setting of implicit neural representation is commonly formulated as learning a mapping function $f_{\boldsymbol{\theta}}$, using a neural network, from an input of low dimensional position coordinates $\mathbf{p} \in \mathbb{R}^d$ to an output value in the target domain. The network training data are coordinates samples as the input and an expected value of the function, or signal, as the output. Examples of such mappings include pixel locations to intensity values for images, or 3D space coordinates to binary surface occupancy labels.

Achieving such mappings using conventional neural networks with ReLU is hard. It has been shown that ReLU networks exhibit a behaviour of *spectral bias* [34, 3, 36, 2, 48, 4]. Networks tend to learn low frequency functions first, as they are characterized by a global behaviour, and are therefore more stable to optimize. Spectral bias, however, also prevents networks from properly learning to fit functions that change at a high rate, e.g., functions with a large Lipschitz constant. When visual domains are concerned, this is mostly evident in delicate details missing from the network output.

To mitigate this deficiency, recent works proposed to replace the ReLU activation layers with periodic sine functions [41] or map the input to some higher dimensional space in order to learn a mapping with high frequencies [30, 43]. In the latter approach, the input $\mathbf{p}$ is encoded to a high dimensional embedding by a family of functionals $e_i : \mathbb{R}^d \to \mathbb{R}$, such that:

$$E(\mathbf{p}) = (e_1(\mathbf{p}), e_2(\mathbf{p}), \ldots, e_n(\mathbf{p})) , \qquad (1)$$

where usually $n \gg d$. Tancik et al. [43], for example, suggest the following encoding:

$$E_{\text{FFN}}(\mathbf{p}) = \left( \cos(2\pi\mathbf{b}_1^\top \mathbf{p}), \sin(2\pi\mathbf{b}_1^\top \mathbf{p}), ..., \cos(2\pi\mathbf{b}_n^\top \mathbf{p}), \sin(2\pi\mathbf{b}_n^\top \mathbf{p}) \right) , \qquad (2)$$

where $\mathbf{b}_i$ are frequency vectors randomly sampled i.i.d. from a Gaussian distribution with standard deviation $\sigma$. Other examples of positional encodings are included in the appendix.

These embedding schemes facilitate *learning* of complex functions, at the price of introducing a second associated phenomenon: the spectral bias characteristic of the network is reduced. The implication in this case is twofold. Positional encodings can cause neural optimizations processes to become unstable, and consequently converge to a bad local minimum (Fig. 9). In addition, when fitting functions of varying levels of details/frequencies, neurons predicting smooth, low frequency areas are still exposed to encoding dimensions of high frequency. That, in turn, may complicate the learning process, as networks have to learn where to ignore such embedding dimensions.

Indeed, Tancik et al. [43] advocate a careful choice of the standard deviation of frequency distribution, $\sigma$ in their method, showing that using low values yields missing details in the network output, and using values that are very high results in noisy artifacts. Thereupon, for such static encodings the value of $\sigma$ requires a parameter sweep per sample.

## 4 Spatially-Adaptive Progressive Encoding (SAPE)

Our proposed approach, SAPE, is a policy for guiding the optimization of implicit neural representations based on input coordinates. SAPE relies on the delicate balance of the two phenomena that govern the learning process: it reconciles the effects of the spectral bias and the expressiveness of high frequency encodings in a manner that benefits from both. It maintains a stable optimization, without sacrificing the ability to fit fine signal details. SAPE is composed of two key components: progressive encoding and spatial adaptivity, which allow it to be less sensitive to the encoding frequency bandwidth, i.e., the choice of the standard deviation in the case of [43] (Fig. 4). Next, we detail each of the mechanisms that compose SAPE. The full algorithm is included in the appendix.

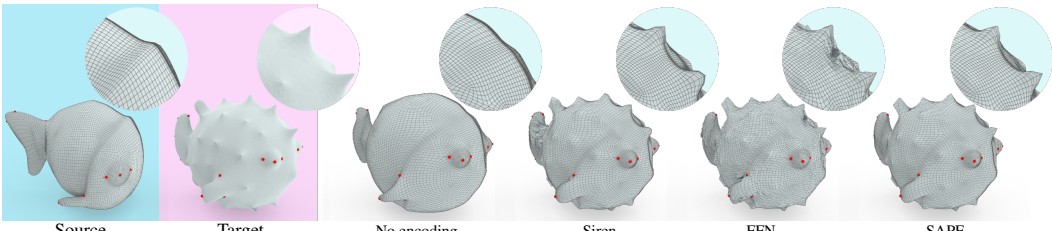

| Source | Target | No encoding | Siren | FFN | SAPE |
|--------|--------|-------------|-------|-----|------|

Figure 5: Mesh transfer. In this task, we transfer the tessellation of a source mesh to a target shape (left columns). We estimate an initial transformation using user specified correspondence points (marked in **red**). Then, we follow with a neural optimization tuning process, which brings the source mesh surface to the target shape while keeping the quality of the original tessellation.

Without loss of generality, in our description below we assume the encoding functionals are sorted by their respective Lipschitz constants ($LC$), i.e. $LC(e_1) \leq LC(e_2) \leq \cdots \leq LC(e_n)$. For example, we sort the Fourier features encoding by the frequency value $\|\mathbf{b}_i\|$. We set the first $d$ dimensions to the identity encodings: $e_i(\mathbf{p}) = p_i$, $\forall i \leq d$, thereby exposing the network to the original input position coordinates $\mathbf{p} \in \mathbb{R}^d$ as well.

**Progressive encoding.** A core part of SAPE is progressively fitting the frequencies. The first layer of the network encodes an input position $\mathbf{p}$ with a set of functionals $e_i$. In order to use only part of the functionals at different stages of training, it multiplies the encoding dimensions with a soft mask, which progresses as a function of the optimization iteration $t$:

$$E_{\text{prog}}(\mathbf{p}, t) = (\widehat{e}_0(\mathbf{p}, t),\ \widehat{e}_1(\mathbf{p}, t),\ \widehat{e}_2(\mathbf{p}, t),\ \ldots,\ \widehat{e}_n(\mathbf{p}, t)) = \boldsymbol{\alpha}(t)^\top E(\mathbf{p}), \tag{3}$$

where $\widehat{e}_i(\mathbf{p}, t) = \alpha_i(t)\, e_i(\mathbf{p})$ is the progression control for the encoding functional $e_i$, regulated by the elements of masking vector $\boldsymbol{\alpha}(t) \in [0,1]^n$. During the optimization, we progressively *reveal* encoding features in a manner that allows the encoding neurons to be tuned to different states according to the value of $\alpha_i(t)$, where $\alpha_i(t) = 1$ corresponds to the state *on*, $\alpha_i(t) = 0$ means *off* and $0 < \alpha_i(t) < 1$ denotes *partially on* (Fig. 3).

We define a policy $\Phi$ that controls the progression of the mask vector $\boldsymbol{\alpha}(t)$ by $\boldsymbol{\alpha}(t + 1) = \Phi\left[\boldsymbol{\alpha}(t)|\boldsymbol{\alpha}(0)\right]$. In this work, we use the following progression rule for the mask of the $i$th encoding dimension at time step $t$:

$$\Phi_P\left[\boldsymbol{\alpha}(t)\right]: \quad \alpha_i(t+1) = \begin{cases} \text{clamp}\left(\dfrac{t - \tau \cdot (i - d)}{\tau},\ 0,\ 1\right), & \text{if } i > d \\ 1, & \text{otherwise}, \end{cases} \tag{4}$$

where $\tau = T/2n$ represents the number of iterations each encoding dimension takes to progress from 0 to 1 and $T$ is the maximal number of iterations in the optimization. Essentially, $\Phi_P$ performs a linear progression sequentially for each encoding $i$ until it is fully exposed, s.t. $\alpha_i(t) = 1$. When a certain encoding dimension mask achieves saturation, we continue to the following one. We allow a technical exception to this rule, for proper progression of correlated masks of encoding dimensions sharing the same Lipschitz constant, e.g., pairs of cos, sin in the encoding in Eq. (2). For initialization, the first $d$ positional encoding masks are fully exposed right from the beginning, and in our setting, correspond with the identity encoding functionals.

Notice that previous methods, which introduce all encoding dimensions to the network at once, can now be formulated as a specific case in our framework, where $\boldsymbol{\alpha}(t + 1) = \mathbf{1}$.

**Spatial adaptivity.** When $t > T/2$, the progression policy $\Phi_P$ achieves *global* saturation in terms of revealing encoding dimensions, so that the masking vector becomes $\boldsymbol{\alpha}(t) = \mathbf{1}$. For signals of global-like characteristics, e.g., have a low $LC$, the network is compelled to learn how to cope with undesirable encoding dimensions of high frequency to regress a smooth, slowly changing signal. This can lead to sub-optimal outputs, often visible in the form of visual artifacts (Fig. 1, mid-bottom).

Given loss function $\mathcal{L}$ and convergence threshold $\varepsilon$, a simple improvement to policy $\Phi_P$ may apply early stopping when $\mathcal{L} < \varepsilon$. However, for signals with varying levels of detail as a function of spatial position $\mathbf{p}$, this improvement is not sufficient: early stopping does not occur due to areas characterized by high frequencies. Relying on a *global* threshold $\varepsilon$ is inherently a sub-optimal decision, as when learning an implicit neural function with positional encoding, different spatial segments of the signal in question may converge at different rates (Fig. 2).

To solve this problem, we set the mask vector to be spatially sensitive: $\boldsymbol{\alpha}(t, \mathbf{p})$. By design, the new policy $\Phi_{SA}$ progresses the encoding of each spatial location $\mathbf{p}$ separately, per encoding dimension $i$. To simplify our explanation, in the current setting we assume the signal can be discretized to distinct spatial segments on a regular grid, each tracked independently (e.g., pixels, voxels). The progressive, spatially adaptive policy is controlled by a feedback loop, where the independent regression loss of each signal segment is used to control the progression:

$$\Phi_{SA}\left[\boldsymbol{\alpha}(t, \mathbf{p})\right]: \quad \boldsymbol{\alpha}(t+1, \mathbf{p}) = \begin{cases} \Phi_P\left(\boldsymbol{\alpha}(t)\right), & \text{if } \mathcal{L}(t, \mathbf{p}) \geq \varepsilon \\ \boldsymbol{\alpha}(t, \mathbf{p}), & \text{otherwise} \end{cases} \tag{5}$$

For implicit functions, at inference time we assume signals can be regressed with "pseudo-continuous" coordinates $\mathbf{p}$. Therefore, to support coordinates $\mathbf{p}$ that did not appear during training and have no mask recordings, we extend the estimated parameters of $\boldsymbol{\alpha}(T, \mathbf{p})$ continuously over the entire input domain by a linear interpolation.

One common example that benefits from spatial adaptivity is natural 2D image containing blurry, out-of-focus areas in the background together with sharp, detailed foreground objects. To shed more light on the behaviour of our algorithm, Figs. 6 and 8 demonstrate examples of spatial heat maps, highlighting the maximal encoding frequency achieved per spatial location, upon convergence.

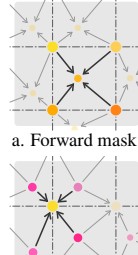

a. Forward mask

b. Loss feedback

**Sparse grid sampling.** We now extend our method to cases where model $f_{\boldsymbol{\theta}}$ is optimized by coordinates $\mathbf{p}$ that do not lie on a regular grid, or when it is simply infeasible to store parameters $\boldsymbol{\alpha}(t, \mathbf{p})$ per sampled coordinate $\mathbf{p}$ during training. In this setting, we discretize the input domain and store parameters $\boldsymbol{\alpha}_{\mathbf{u}}$ in a *sparse* grid $G$, where for a grid of resolution $\mathbf{r}^d \subseteq \mathbb{R}^d$, we denote $\mathbf{u}$ as the multidimensional grid coordinates. In the forward pass, we obtain the original encoding mask $\boldsymbol{\alpha}(t, \mathbf{p})$ per sampled point $\mathbf{p}$ by encoding a linear interpolation of the masking parameters over the nearest grid nodes (see the inset (a) on the left):

$$\boldsymbol{\alpha}(t, \mathbf{p}) = \sum_{\mathbf{u} \in \mathcal{N}_G(\mathbf{p})} w_{p,\mathbf{u}} \boldsymbol{\alpha}(t, \mathbf{u}), \tag{6}$$

where $\mathcal{N}_G(\mathbf{p})$ are the neighboring grid coordinates in the vicinity of $\mathbf{p}$, and $w_{\mathbf{p},\mathbf{u}}$ are the interpolation weights for a sample $\mathbf{p}$ over the multidimensional grid $G$ such that $\sum_{\mathbf{u} \in \mathcal{N}_G} w_{\mathbf{p},\mathbf{u}} = 1$. During training, the loss $\mathcal{L}(t, \mathbf{u})$ at time $t$ for a given grid point $\mathbf{u}$ is accumulated over all sampled points $\mathbf{p}$ affected by $\mathbf{u}$ (inset (b) above):

$$\mathcal{L}(t, \mathbf{u}) = \frac{1}{\sum_{\mathbf{p}} w_{\mathbf{p},\mathbf{u}}} \sum_{\mathbf{p}} w_{\mathbf{p},\mathbf{u}} \mathcal{L}(t, \mathbf{p}). \tag{7}$$

Here, $\mathcal{L}(t, \mathbf{p})$ is the loss for the original coordinate $\mathbf{p}$. To obtain the overall training loss, we sum over all the used points for training at time $t$. We note that the linear interpolation used during the forward pass has an added benefit: since only local neighbors participate per update of the original coordinate $\mathbf{p}$, sparse weights allow for efficient accumulations during forward and backward pass updates.

## 5 Experiments

We evaluate SAPE on a variety of common 2D and 3D regressions tasks. In addition, we demonstrate how it may be used to improve the tasks of deforming 2D silhouettes and transferring mesh connectivity between 3D shapes. We demonstrate our results using the Fourier feature encoding [43] (Eq. (2)). We emphasize that SAPE is agnostic to the encoding used and applicable to other mapping schemes as well. More examples, as well as the full implementation details appear in the appendix.

### 5.1 Evaluations

We test SAPE on a variety of problems: regression tasks optimized by a direct supervision and geometric tasks optimized by an indirect supervision. All configurations employ 256 unique frequency encodings sampled from a Gaussian distribution. For fair evaluation, the distribution scale and number of neural layers remain the same for all inputs in the same problem setting. However, hyperparameter values may vary per problem. For convergence threshold $\varepsilon$, we set the values of $1e-3$ for regression tasks and $1e-2$ for geometric tasks. See the appendixfor full description of implementation details. We compare the settings of 6 MLP configurations:

Table 1: Quantitative Evaluation of SAPE against baseline encoded MLPs on various tasks. Best results displayed in **bold**. $\pm$ accounts for deviation among independent experiments.

| | 2D regression (PSNR) | | 3D occupancy (IoU) | | 2D silhouettes (IoU) |
|---|---|---|---|---|---|
| | **Natural images** | **Text images** | **Thingi10K** | **Turbosquid** | **MPEG7 (IoU)** |
| No Encoding | $19.72 \pm 2.77$ | $19.39 \pm 2.02$ | $0.899 \pm 0.034$ | $0.872 \pm 0.08$ | $0.81 \pm 0.177$ |
| SIREN | $27.03 \pm 4.28$ | $30.81 \pm 1.72$ | $0.964 \pm 0.013$ | $0.93 \pm 0.025$ | $0.779 \pm 0.253$ |
| RBFG | $25.98 \pm 3.98$ | $25.94 \pm 4.03$ | $0.941 \pm 0.022$ | $0.919 \pm 0.044$ | $0.854 \pm 0.155$ |
| SAPE + RBFG | $27.06 \pm 3.94$ | $28.52 \pm 3.51$ | $0.966 \pm 0.015$ | $0.94 \pm 0.03$ | $0.928 \pm 0.102$ |
| FF | $25.57 \pm 4.19$ | $30.47 \pm 2.11$ | $0.945 \pm 0.03$ | $0.964 \pm 0.017$ | $0.873 \pm 0.142$ |
| Progressive FF | $27.01 \pm 3.56$ | $30.2 \pm 1.83$ | $0.98 \pm 0.018$ | $0.943 \pm 0.019$ | $0.86 \pm 0.133$ |
| SAPE + FF | $\mathbf{28.09 \pm 4.04}$ | $\mathbf{31.84 \pm 2.15}$ | $\mathbf{0.981 \pm 0.008}$ | $\mathbf{0.969 \pm 0.013}$ | $\mathbf{0.928 \pm 0.095}$ |

1) *No encoding*: Basic ReLU MLP without encoding. 2) *SIREN*: An MLP with sine activations based on the implementation of Sitzmann et al. [41]. 3) *RBF-grid*: An MLP with repeated radial basis function as a first encoding layer. 4) *SAPE + RBF-grid*. 5) *FFN*: An MLP network with Fourier features as the first encoding layer, see Eq. (2). 6) *SAPE + FFN*.

Table 2: Evaluation on the mesh transfer task.

| | Chamfer ($\downarrow$) | Hausdorff ($\downarrow$) | Dirichlet ($\downarrow$) |
|---|---|---|---|
| No Encoding | 1.37 | 1.36 | **1.3** |
| SIREN | 0.74 | 1.32 | 4.75 |
| RBFG | 0.97 | 0.87 | 1.65 |
| SAPE + RBFG | 0.55 | 0.71 | 1.65 |
| FF | 1.27 | 0.94 | 2.22 |
| SAPE + FF | **0.39** | **0.49** | 1.82 |

The bandwidths of encoding functions in 3) and 5) are optimally selected by a grid search over a validation set or taken from a public implementation, depending on the task. For the SAPE variants we use double the $\sigma$ of these bandwidths, as SAPE is not sensitive to a particular value, as long as it allows high frequency encodings. The quantitative results are summarized in Tables 1 and 2. Below is an overview of each task.

**2D image regression**. In this task we optimize the networks to map 2D input pixel coordinates, normalized to $[-1, 1]^2$, to their RGB values. We conduct the evaluation on the same test sets as Tancik et al. [43], which contain a dataset of natural images and a synthetic dataset of text images. Similar to them, the network is trained on regularly spaced grid containing $25\%$ of the pixels. We use the evaluation metric of PSNR over the entire image, compared to the ground truth image. Quantitative results are reported in Table 1 on the left.

**3D occupancy.** In this task, we use a similar setting to occupancy networks [28]: the model is trained to classify an input 3D coordinate for being inside or outside the training shape. For training, we sample 9 million points divided into 3 equal groups: uniformly sampled points in $[-1, 1]^3$, surface points perturbed with random Gaussian noise vectors using $\sigma = 0.1$ and $\sigma = 0.01$. We evaluate the quality of the result by estimating the intersection-over-union (IoU) with respect to the ground truth shape by sampling additional $1e6$ random points and $1e6$ more challenging samples near the surface, and report the average IoU score.

We compare the networks on two test sets. The first set is composed of 10 selected models from the Thingi10K dataset [50]. The second, more challenging test set is composed of 4 models from TurboSquid[2]. The quantitative results are summarized in Table 1, middle section. Qualitative results are shown in the appendix.

**2D silhouettes.** Here, we optimize an MLP to deform a unit circle represented as a polyline to a target 2D point cloud of a silhouette shape. Fig. 9 shows a number of examples of such target shapes. We start by *calibrating* the MLP to learn a simple mapping from $p \in [0, 1]$ to the 2D unit circle: $f(p) = (\cos(2\pi p), \sin(2\pi p))$. We then optimize the mapping function by minimizing the symmetric chamfer distance between the network output and the silhouette.

To evaluate the performance of the different methods, we test the networks on 20 shapes from the MPEF7 dataset [20] and measure the intersection over union between the resulting shape and the target shape; the results are reported in Table 1. Fig. 9 shows qualitative results of this task and snapshots from the optimization process. Due to the expressiveness of both FFN and SIREN networks, the chamfer loss causes distortions in the early stage of the optimization that cannot be recovered, leading to undesired results. The coarse-to-fine optimization of SAPE allows both avoiding distortions and matching high frequencies of the target shape.

---

[2]`https://www.turbosquid.com`

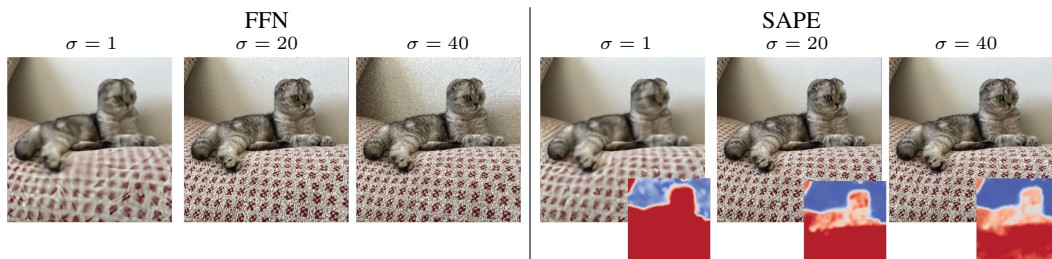

Figure 6: 2D Image regression comparing different distributions of bandwidths for Fourier Frequencies. SAPE is more robust to the choice of $\sigma$, consistently producing pleasing results in terms of both PSNR and visual quality.

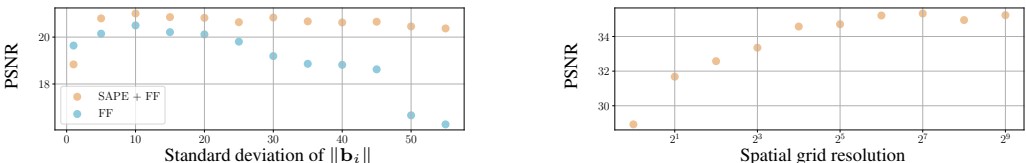

Figure 7: Left: PSNR as a function of the Fourier features' standard deviation of the encoding for the example in Fig. 6. Right: PSNR as a function of spatial grid resolution for the example in Fig. 8.

**3D mesh transfer.** In this task, we would like to transfer a 3D source mesh to a target shape [45, 9, 12, 38, 39], which is represented by a mesh or a point cloud. The MLP receives the vertices of the source mesh and outputs transformed vertices, such that together with the source tessellation, the optimized mesh fits the target shape while respecting the structure of the source mesh in terms of distortion. In addition, we may utilize a set of marked correspondence points between the input shapes that enable the estimation of an initial affine transformation between the source and the target, followed by a biharmonic deformation [17]. The optimization loss for this task is composed of two terms: A distance loss that measures the symmetric chamfer distance between the optimized mesh and the target shape, and a structural term that measures the discrete conformal energy between the optimized mesh and the source mesh.

We test the different networks on 6 pairs of meshes and evaluate the results by measuring the chamfer and Hausdorff distances between the target shape and the output mesh. The distortion of the transferred meshing is measured by Dirichlet energy with respect to the source mesh. Table 2 shows the quantitative results. Qualitative results are shown in Fig. 5.

Standard MLPs struggle to fit the target shape and remain close to the source mesh, therefore obtaining low Dirichlet energy but also high Chamfer and Hausdorff distances and thus failing the task. SAPE's feedback specifies the regions on the mesh that are distanced from the target shape. That allows the optimization to gradually increase the frequencies used in high curvature areas while avoiding large distortions at the beginning of the optimization, when global deformations take place. For that reason, contrary to other methods such as FFN and SIREN, SAPE is able to avoid solutions of bad local minima and reconstruct the target signal better.

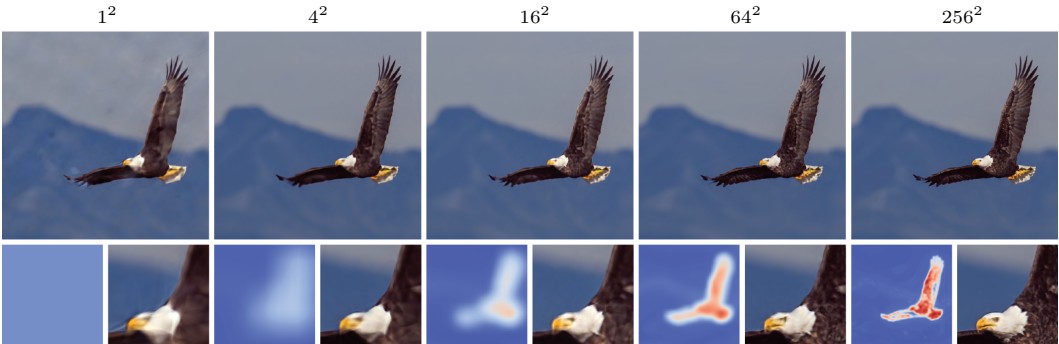

Figure 8: Influence of the spatial grid resolution on output quality. Top row: Output of SAPE with grid resolutions of $1^2$, $4^2$, $16^2$, $64^2$ and $256^2$. Bottom left: Interpolated heatmap of maximal frequency unmasked per grid position. Bottom right: Enlarged results of area with a range of frequencies.

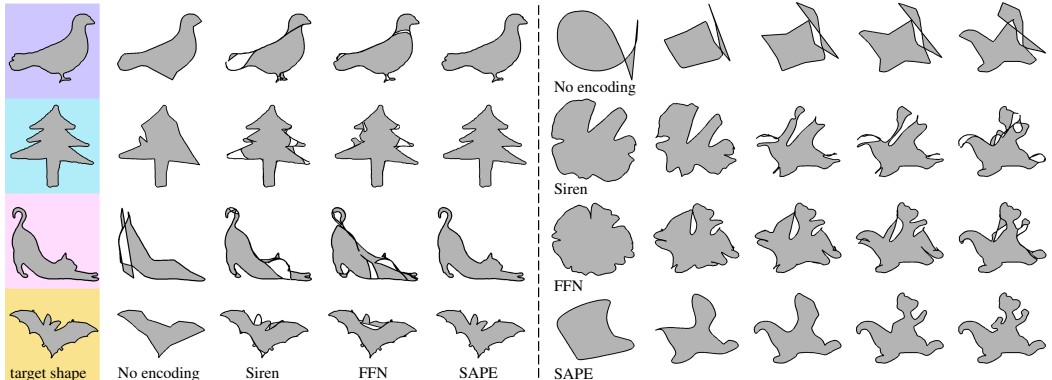

Figure 9: Iterative deformation of 2D silhouettes. Left: comparison of different network configurations when optimizing for the deformation of a unit circle to the target shape, represented by a polyline. Right: snapshots of the optimization process. Notice that SAPE (bottom) avoids large distortion at the beginning of the optimization, and can therefore fit the delicate details of the target shape as the optimization progresses.

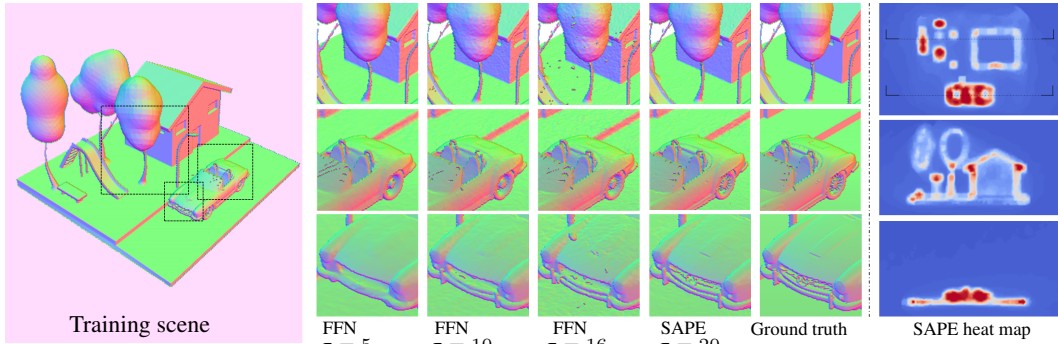

Figure 10: 3D occupancy scene regression. By spatially adapting the volumetric encoding for the occupancy network, SAPE can fit small details such as the car wheels without introducing noise in smooth or empty regions. To the right, we display cross sections of the 3D heat map of SAPE. Notice that smooth surfaces, like the the trees and the roof, utilize lower frequency encodings compared to detailed objects like the car and the slide.

## 5.2 Ablation

**Progressive vs. non-progressive.** Similar to the setting described in Section 5.1, we train an MLP on randomly sampled pixels of images. We map 2D pixel coordinates to RGB values and measure the PSNR over all pixels compared to the ground truth image.

Fig. 6 and Fig. 7 (left) show a comparison between the performance of FFN with and without SAPE when training on 25% of the pixels. Our results show that due to progressive-spatial adjustment of the encoding, SAPE is less sensitive to tuning of the standard deviation ($\sigma$) of distribution of frequencies $\mathbf{b}_i$. By contrast, FFN suffers from a fragile trade-off between underfitting and overfitting, which results in either blurry pixels in high frequency regions or noise in the smooth background areas.

A similar phenomena is shown in the 3D example in Fig. 10. In this task, we train an occupancy network using the setting described in Section 5.1. For regulating the level of encoding across the scene our method uses a voxel map of resolution $128^3$ for spatial encoding. This map is updated based on feedback loss during training. Without SAPE, low frequency encodings cannot represent detailed structures like the car object in the scene. Increasing the frequency bandwidth results in noisy surfaces and undesired blob artifacts in empty spaces. By contrast, SAPE achieves better representation of detailed regions, as well as of smooth surfaces.

**Spatial progressive vs. non-spatial progressive.** To demonstrate the advantage of using spatially adaptive encoding, we compare it to a variant of SAPE that *globally* progresses the frequency level of encodings for all spatial areas equally, and converges when all training samples fit. Fig. 2 shows the advantage of using the spatial encoding in SAPE for a 1D regression task. In addition, Table 1

shows quantitative improvement gained by SAPE applied on FF, compared to a *global* progression (Progressive FF).

Note that the non-spatial variant is equivalent to SAPE with a spatial grid of resolution one. Fig. 8 and Fig. 7 (right) show the benefit of the spatial encoding component for a 2D image regression task (similar to the one above). We also show how increasing the grid resolution affects the quality of the result in a 2D image regression task. We observe that the quality stays at its highest when the spatial grid resolution approximates the sampling ratio of coordinates during training.

## 6    Limitations and Future Work

When SAPE is employed with a sampling grid, the chosen resolution provides a trade-off between memory and quality. Choosing a grid of low resolution may yield sub-optimal outputs (see examples in the appendix). As SAPE's varying frequencies sampling grid bears some resemblance to multi-resolution analysis and wavelet transforms, future extensions to SAPE may solve this issue by maintaining the masking parameters in a sparse structure such as an Octree [42] or GMM [10].

When using very high frequencies, SAPE may enhance inherent output noise, as the maximal frequency encoding gets picked for the noisy portions of the signal.

In this work we investigated a linear progression policy for unmasking frequency encodings. Followup works may extend our scheme to a more optimized curriculum, for example, using meta-learning.

## 7    Conclusions

We presented a policy for improving the quality of signals learned by coordinate based optimizations. We rely on two major contributions: progressive introduction of positional encodings, and spatial adaptivity, which allows different rates of progression per signal location. Our method is simple to implement and improves the implicit neural representation result of MLP networks in various tasks.

Given the surge of applications that use positional encodings, and the ease of deploying SAPE, we believe our approach will be useful for many problems in the vision and graphics domains. Specifically, the fact that SAPE is encoding-agnostic and insensitive to most encoding hyperparameters provides stable training and facilitates the use of positional encodings in novel tasks.

## 8    Broader Impact Statement

SAPE is a method for boosting the accuracy of learned implicit neural functions. It can be leveraged to enable new applications or improve various downstream tasks that involve precise processing of signals, like segmentation, or reconstruction of complex signals, such as streaming video or signed distanced functions. Beyond the domains of vision and graphics, we conjecture SAPE might also benefit applications in speech, audio, and signal processing in general. Depending on their use, all mentioned applications may have a positive or negative impact. SAPE can be used to facilitate novel applications that require accurate recovery of signals, like 3D reconstruction for AR and VR. It can also be misused for the purpose of generating fake media, including audio, video and images.

## Acknowledgements

We would like to thank Hao (Richard) Zhang, Hadar Averbuch-Elor and Rinon Gal for their insightful comments, and the anonymous reviewers for their helpful remarks. This work was supported in part by the Israel Science Foundation grant no. 2492/20 and by the ERC-StG grant no. 757497 (SPADE).

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
