# SAPE – Supplementary Materials

**Amir Hertz**    **Or Perel**    **Raja Giryes**    **Olga Sorkine-Hornung**    **Daniel Cohen-Or**

## Contents

35th Conference on Neural Information Processing Systems (NeurIPS 2021).

# 1 Encoding Types

In this section we describe the formulation of additional positional encodings. The input to the encoding function is a position $\mathbf{p} \in \mathbb{R}^d$ and the width of the encoding is specified by a hyperparameter $n$. The used encoding may have an additional hyperparameter $\sigma$ that determines the frequencies of the encoding (which also sets the Lipschitz constant of the encoding). A $2D$ example for the described encoding types is shown in Fig. 1

**Radial Basis Functions (RBF)** [5] ($E_{RBF} : R^d \longrightarrow R^n$)

$$E_{\mathrm{RBF}}\left(\mathbf{p}\right) = \left[\exp\left(-\frac{\|\Delta_{p,i}\|^2}{b_i^2}\right)\right] \text{ for } 1 \le i \le n, \tag{1}$$

where $\Delta_{p,i} = \mathbf{p} - \mathbf{c}_i$, $\mathbf{b} \in \mathbb{R}^n$ and $\mathbf{c}_i \in \mathbb{R}^d$. The parameters $\mathbf{c}_i$ are the centers of each encoding and are uniformly sampled i.i.d. from $[-1, 1]^d$. The value of $1/b_i$ (where $b_i$ is the $i$th entry of $\mathbf{b}$) is randomly sampled i.i.d. from a uniform distribution with a standard deviation $\sigma$. Notice that the Lipschitz constant of the RBF encoding grows linearly with $\frac{1}{b_i}$.

**Periodic Radial Basis Functions** ($E_{PRBF} : R^d \longrightarrow R^{2n}$) As demonstrated by Sitzmann et al. [6] MLP with a RBF encoding layer struggles to represent high frequencies functions. However, we have found that we can improve the expressiveness of the RBF by making it periodic. We use the same formula as the one of regular RBF but with a modification of the formulation of $\Delta_{p,i}$:

$$\Delta_{p,i} = 2\left((\mathbf{p} - \mathbf{c}_i) \bmod 2b_i\right) - 2b_i, 2\left((\mathbf{p} - \mathbf{c}_i + b_i) \bmod 2b_i\right) - 2b_i, \tag{2}$$

where $\mathbf{b} \in \mathbb{R}^m$ and $\mathbf{c}_i \in \mathbb{R}^d$ are randomly sampled as in the RBF case above.

To get an intuition to the improvement notice that (the formula holds both for RBF and PRBF):

$$\frac{\partial E_{RBF}}{\partial \Delta_{p,i}} = -\frac{2\Delta_{p,i}}{b_i^2}\exp\left(-\frac{\|\Delta_{p,i}\|^2}{b_i^2}\right).$$

Since $\left\|\frac{\partial E_{RBF}}{\partial \Delta_{p,i}}\right\| \xrightarrow[\|\Delta_{p,i}\| \longrightarrow \infty]{} 0$, each RBF encoding can only represent high frequency in a single location. Making $\Delta_{p,i}$ periodic prevents this vanishing of the gradient to zero and therefore allows the PRBF to represent high frequencies in multiple locations and not only at the center (where $\Delta_{p,i}$ is low for the regular RBF). Thus, it facilitate learning high frequency mapping across the entire domain of $\mathbf{p}$. Due to this advantage of PRBF, we use it in the experiments as it leads to better encoding.

**Fourier Features** [7] ($E_{FFN} : R^d \longrightarrow R^{2n}$) As shown in the papar, the FF encoding is given by

$$E_{\mathrm{FFN}}\left(\mathbf{p}\right) = [\cos(2\pi\mathbf{b}_1^\top\mathbf{p}), \sin(2\pi\mathbf{b}_1^\top\mathbf{p}), ..., \cos(2\pi\mathbf{b}_n^\top\mathbf{p}), \sin(2\pi\mathbf{b}_n\mathbf{p})]^\top, \tag{3}$$

where $\mathbf{b}_i \in \mathbb{R}^d$ are randomly sampled i.i.d. from a Gaussian distribution with standard deviation $\sigma$. Similar to the RBF encoding, the Lipschitz constant of FF encoding grows linearly with $\|\mathbf{b}_i\|$.

**"Regular Positional Encoding"** [4] ($E_{PE} : R^d \longrightarrow R^{2n}$)

$$E_{\mathrm{PE}}\left(\mathbf{p}\right) = [\cos(2^i\pi\mathbf{p}_j), \sin(2^i\pi\mathbf{p}_j)]^\top \text{ for } 0 \le i < n, 1 \le j \le d$$

As can be seen, "Regular Positional Encoding" is a unique case of $FF$ where the vectors $b_i$ are parallel to the axes and their magnitude grows exponentially. In our experiments, we use the more general FF encoding.

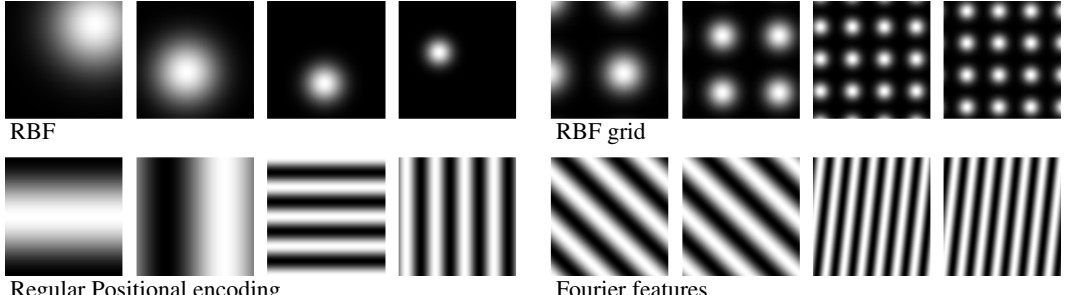

RBF                RBF grid

Regular Positional encoding       Fourier features

Figure 1: Encoding types. In order to learn a high frequency function, the input to the network, $\mathbf{p}$, is encoded to a high dimensional space by a family of encoding functions $E$ characterize by increasing size of Lipschitz constants.

## 2 Algorithms

The SAPE algorithm is summarized as pseudo-code below. See Section 4 in the paper for a detailed description of the motivation behind the various modules in the algorithm.

---

**Algorithm 1:** SAPE Algorithm

---

**Hyperparameters:** Coordinates mapping function $E : \mathbb{R}^d \mapsto \mathbb{R}^n$,
Encoding-weights grid resolution $g_r{}^d$, Weight progression policy $\Phi$, Convergence threshold $\varepsilon$
**Input:** Samples of coordinates $\mathbf{p} \in \mathbb{R}^d$, Corresponding signal values $\mathbf{y} \in \mathbb{N}$.

1     randomly initialize network weights $\boldsymbol{\theta}_0$.
2     initialize encoding weights $\boldsymbol{\alpha}_0 \in g_r{}^d$.
3     **for** *training step* $t \in T$ **do**
        /* Obtain encoding mask per interpolated grid weights by $\mathbf{p}$    */
4        $\boldsymbol{\alpha}_t[\mathbf{p}] \longleftarrow interpolate_{\boldsymbol{\alpha}_t}(\mathbf{p})$
5        $\mathbf{x}_t \longleftarrow \boldsymbol{\alpha}_t[\mathbf{p}]^\top E(\mathbf{p})$ /* Map coordinates to network input      */
6        $\hat{\mathbf{y}}_t \longleftarrow f_\theta(\mathbf{x}_t)$ /* Obtain network prediction, per coordinate $\mathbf{p}$    */
7        **for** *grid coordinate* $g_p \in g_r{}^d$ **do**
           /* If weighted grid coordinate loss is above threshold    */
8           **if** $\mathcal{L}(\hat{\mathbf{y}}_t, \mathbf{y}) \geq \varepsilon$ **then**
             /* Increase weight for encoding frequencies      */
9             $\boldsymbol{\alpha}_{t+1}[\mathbf{p}] \longleftarrow \Phi(\boldsymbol{\alpha}_t[\mathbf{p}])$
10          **end**
11        **end**
12        $\boldsymbol{\theta}_{t+1} \longleftarrow \boldsymbol{\theta}_t - \beta \nabla_{\boldsymbol{\theta}_t} \mathcal{L}(\hat{\mathbf{y}}_t, \mathbf{y})$
13     **end**

---

# 3 Additional Experiments

We show here additional experiments of the image regression task.

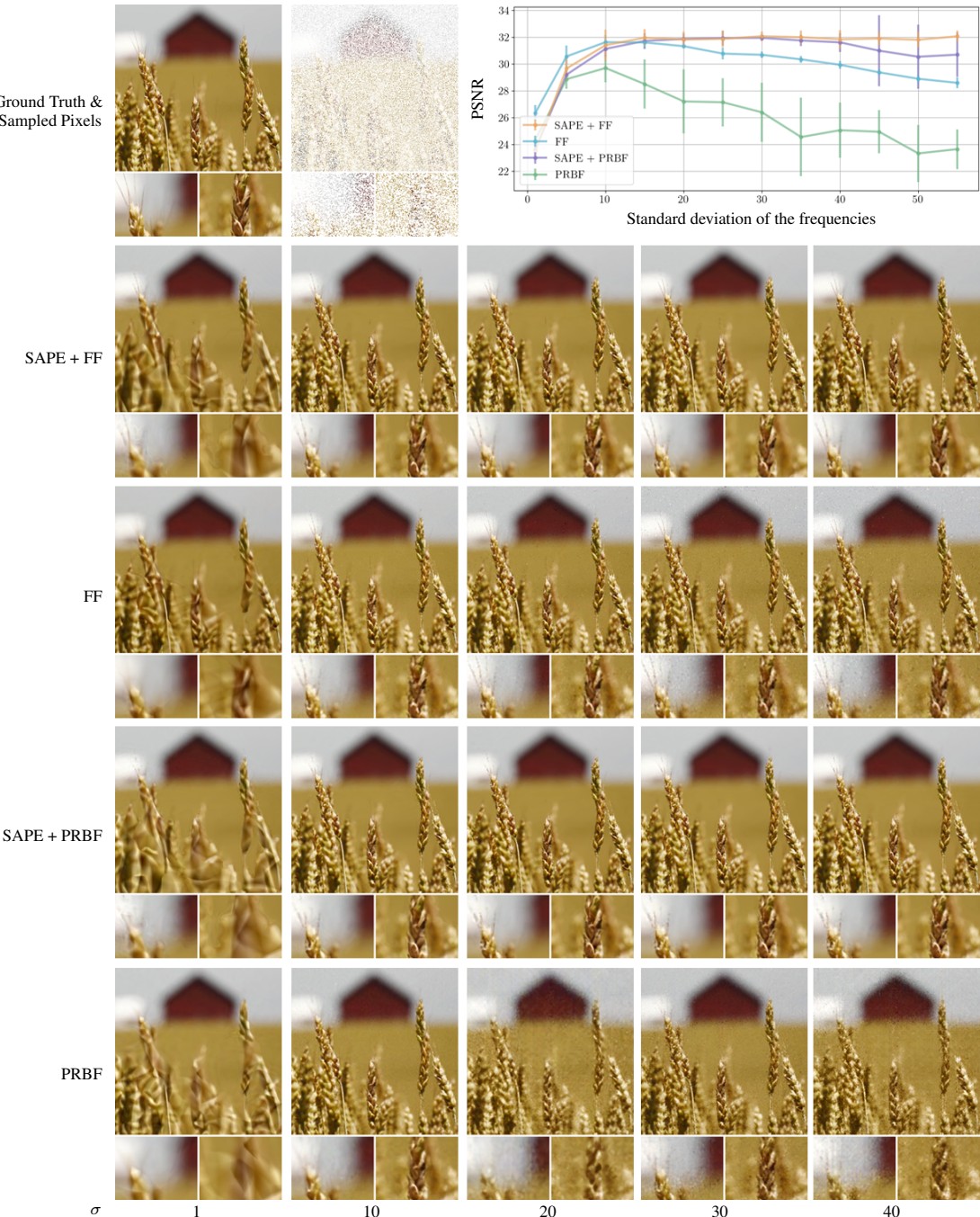

Figure 2: 2D image regression with an increasing size of the standard deviation of the encoding frequency.

**Robustness to the frequency distribution.** We show in Figure 2 another example to the one in the paper where we evaluate the implicit image regression task with increasing size of the standard deviation $\sigma$ of the encoding frequency distribution– the distribution of $\|b_i\|$ for the Fourier Features encoding (3) and the distribution of $1/b_i$ for the Periodic RBF (1).

In this example, we train the networks on 25% of the pixels and measure the PSNR of the whole image. We show the mean score and error bars of 10 experiments for each network configuration and a selection of $\sigma$.

**Robustness to sample size.** In this experiment we test the quality of the reconstruction on different values of sample rates– number of pixels we train the network on. Qualitative results are shown in Figure 3 and quantitative results are shown in Figure 4.

In this example, we use $\sigma = 20$ for the SAPE networks and $\sigma = 15$ for FF and PRBF networks. We show the mean PSNR score and error bars of 10 experiments for each network configuration and a selection of sample rate.

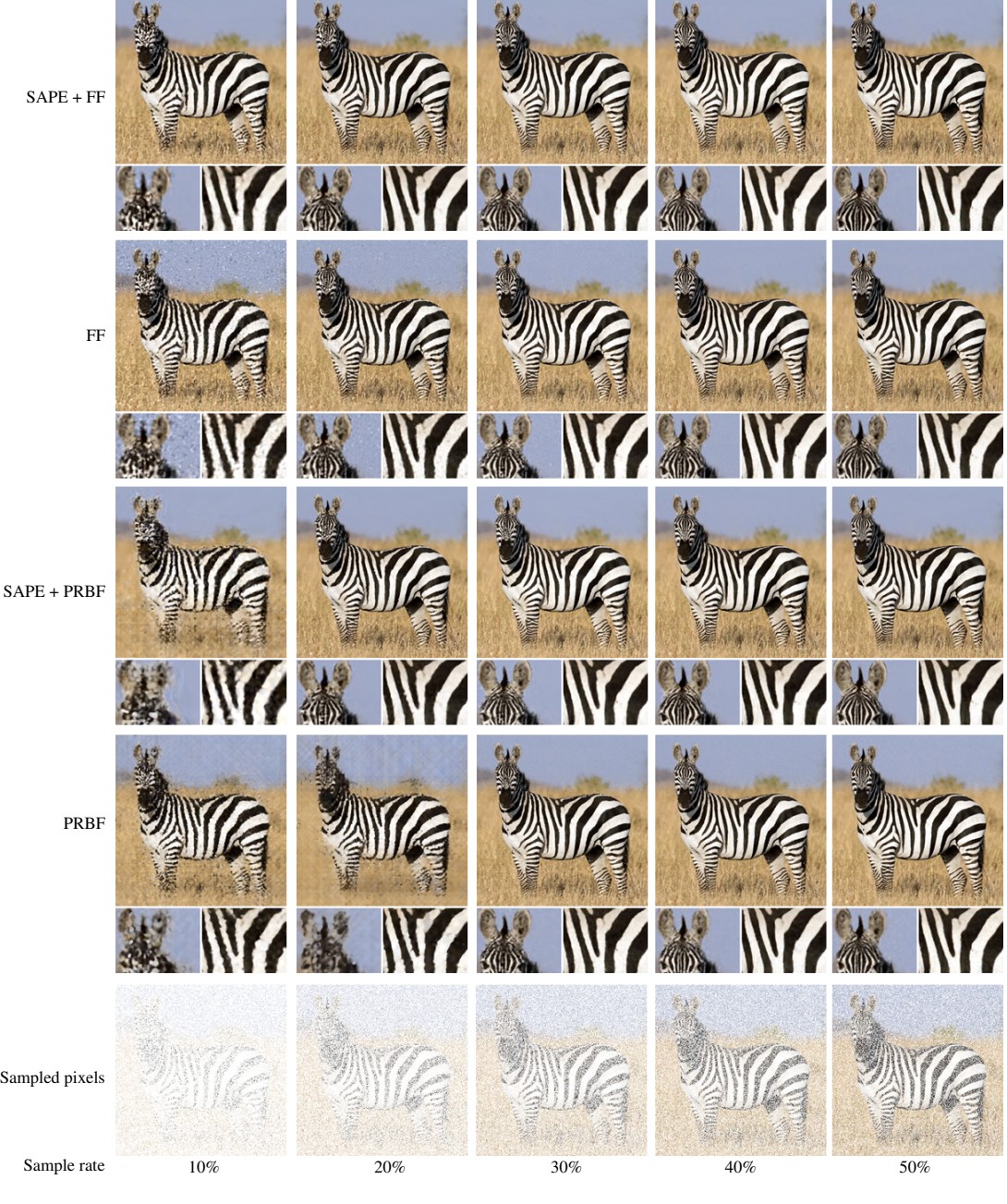

Figure 3: 2D image regression with increasing size of the training pixels sample rate.

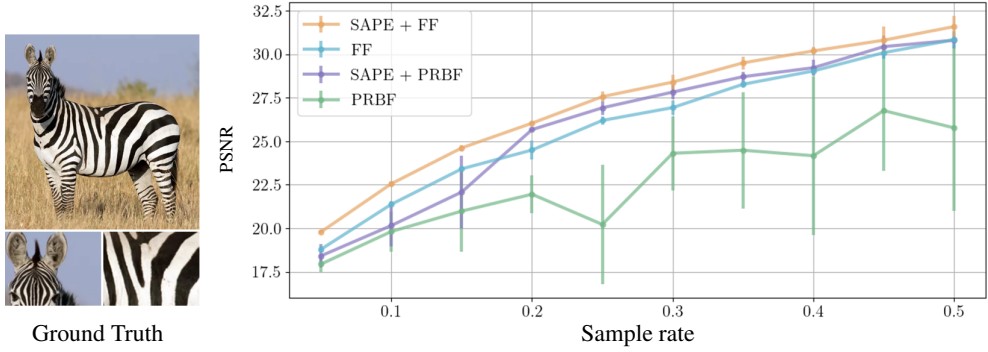

Ground Truth

Figure 4: 2D image regression with increasing size of the standard deviation – encoding frequency.

# 4 Implementation Details

We describe below the network configuration and hyperparameters we used in our experiments.

We report the run times of the optimizations on a single GeForce GTX TITAN X GPU with 12GB memory size.

## 4.1 1D Signals Fitting

In the 1D example (Figure 2 in the paper) we train the networks to fit a 1D function by fitting 26 points sampled from the underlying signal. The different networks contains 2 hidden layers of size 32. The Fourier Features configuration is composed of 256 frequencies randomly sampled from a Gaussian distribution with standard deviation with $\sigma = 3$. Each model is trained for 10000 iterations using the Adam [2] optimizer with default settings ($\beta_1 = 0.9, \beta_2 = 0.999, \varepsilon = 10^{-8}$). We use a learning rate of $10^{-5}$ for the encoding based networks. Due to slow convergence, we use a learning rate of $10^{-4}$ for the baseline configuration ("No Encoding"). Each optimization took about 20 seconds.

## 4.2 Image Regression

For the ablation experiments and the evaluations (Table 1 in the paper, left) we trained the networks (MLP, 3 hidden layers with 256 channels) for 5000 iterations. The Fourier Features and periodic RBF were trained with 256 unique frequency levels.

Unless specified otherwise, for Fourier Features, we use the reported configurations in [7]: $\sigma = 10$ for the natural images dataset and $\sigma = 14$ for the text dataset. For the Periodic RBF we chose $\sigma$ by evaluations on the provided hold-out set. We found that $\sigma = 14$ works best. Unless specified otherwise, for SAPE, we use $\sigma = 20$ and an encoding masking grid of size $128 \times 128$.

Each optimization took about 5 minutes.

Additional qualitative results for this task are shown in Figure 5.

## 4.3 3D Occupancy

In the 3D Occupancy evaluations (Table 1 in the paper, middle) we trained the networks to classify an input 3D coordinate for residing inside or outside the training shape.

We train the networks (MLP, 4 hidden layers with 256 channels and a sigmoid at the output) for 50 epochs on the 9 million sampled points with batch size of 5000. The Fourier Features and periodic RBF were trained with 256 frequencies.

We used $\sigma = 12$, as reported in the official implementation of FFN [7] and $\sigma = 16$ for the Periodic RBF, selected by running on a hold-out example. For SAPE, we used $\sigma = 20$. For the encoding

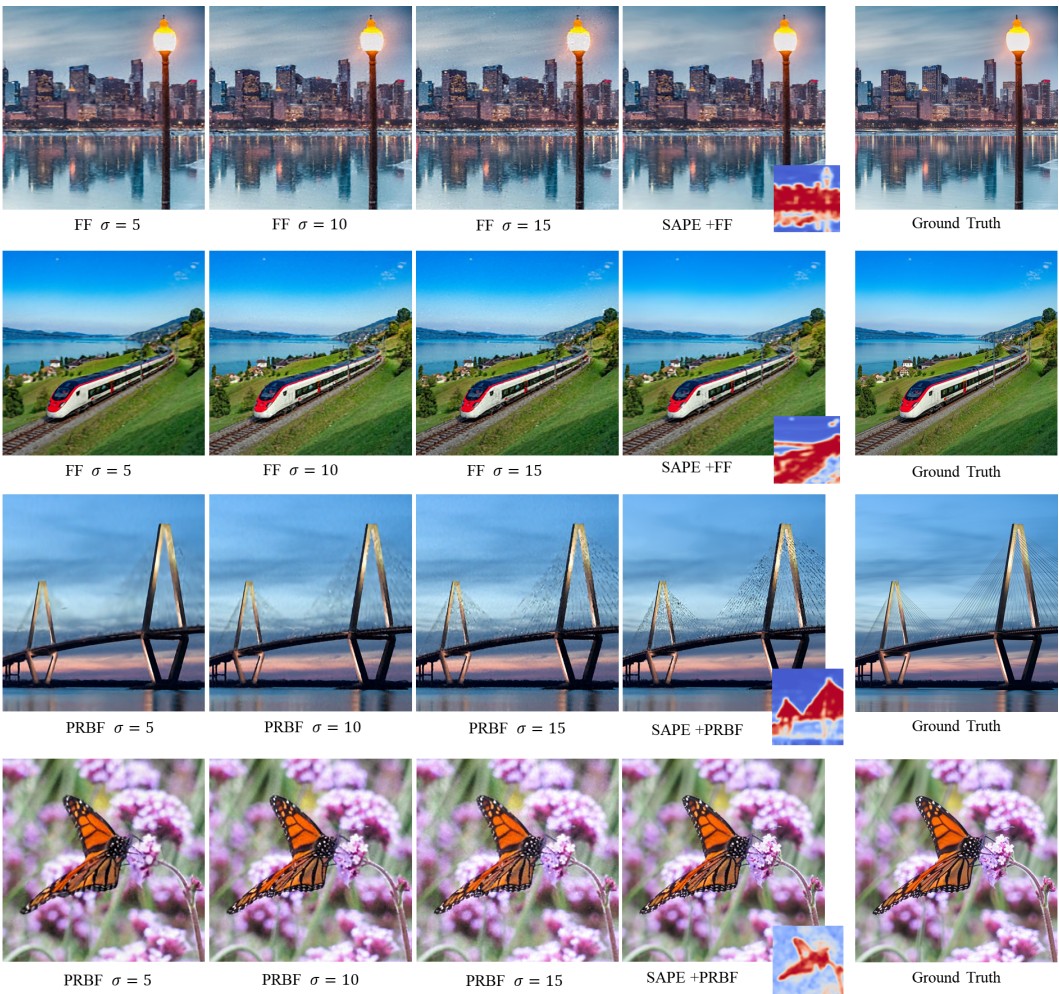

Figure 5: Additional 2D regression comparisons. THe network traind on 25% of the pixesl.

masking grid, we used a grid size of $64^3$ for the Thingi10k [8] evaluation and for the more challenging meshes from TurbuSquid[1] we used grid size of $128 \times 128$.

Each model was trained using the Adam optimizer with the default settings ($\beta_1 = 0.9, \beta_2 = 0.999, \varepsilon = 10^{-8}$) and learning rate of $10^{-4}$.

Each optimization took about 1 hour.

### 4.4 2D Silhouettes Deformation

In this evaluation (Table 1 in the paper, right) we trained the networks to deform a unit circle represented as a polyline to a target 2D point cloud of a silhouette shape.

We trained the networks (MLP, 3 hidden layers with 256 channels) for 10000 iterations. The Fourier Features and periodic RBF were trained with 256 frequencies.

We used $\sigma = 4$ for both FF and Periodic RBF encoding distributions which were selected by grid search on a hold-out set. For SAPE, we used $\sigma = 8$.

Each model was trained using the Adam optimizer with its default settings ($\beta_1 = 0.9, \beta_2 = 0.999, \varepsilon = 10^{-8}$) and learning rate of $10^{-5}$ for the encoded networks, $10^{-4}$ for the base MLP and $10^{-6}$ for SIREN since it may become unstable with higher *lr*. Each optimization took about 2 minutes.

---

[1] https://www.turbosquid.com

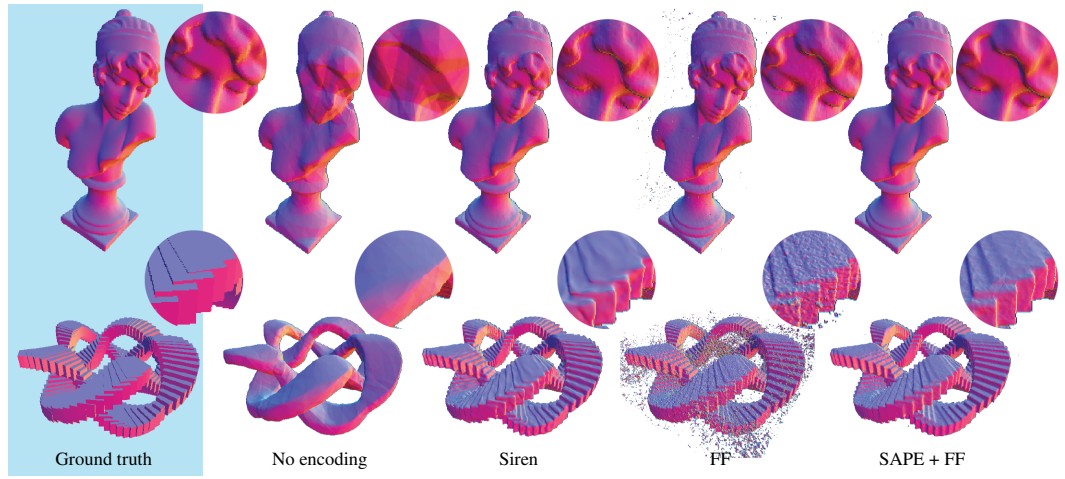

Figure 6: 3D Occupancy comparison on selected meshes from Thingi10k [8] dataset.

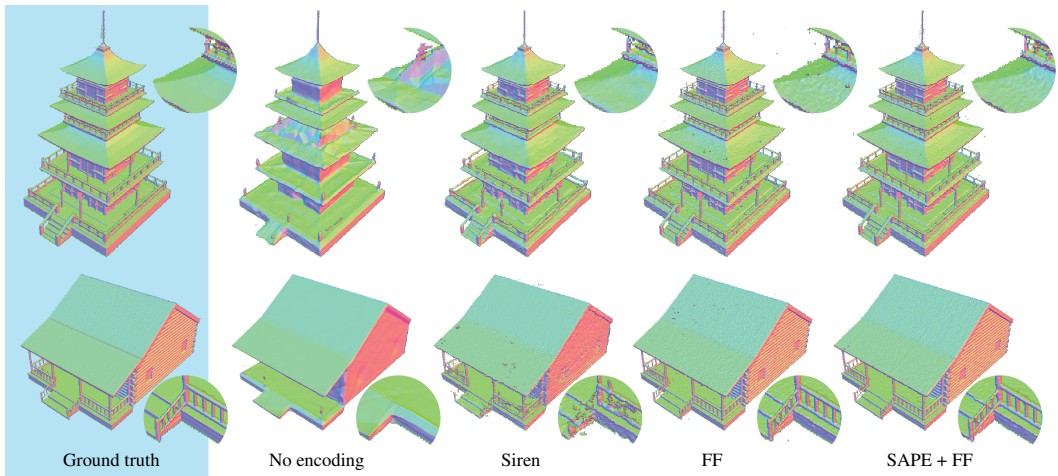

Figure 7: 3D Occupancy comparison on selected meshes from TurboSquid.

### 4.5 3D Mesh Transfer

In this evaluation (Table 2 in the paper) we trained the networks to transfer a 3D source mesh to a target shape.

We trained the networks (MLP, 4 hidden layers 256 channels) for 10000 iterations. The Fourier Features and periodic RBF were trained with 256 frequencies.

We used $\sigma = 3$ for both FF and Periodic RBF encoding distributions which were selected by a grid search on a hold-out set. For SAPE, we used $\sigma = 6$.

Each model was trained using the Adam optimizer with its default settings ($\beta_1 = 0.9, \beta_2 = 0.999, \varepsilon = 10^{-8}$) and learning rate of $10^{-4}$.

Additional overview for this task optimization is given in Section 5 of this appendix. Additional qualitative results for this task can be seen in Figure 8.

An optimization process for this task takes about 5 minutes.

As shown in Figure 9, in the 2D silhouette and mesh transfer settings, our method may result with distorted output when dealing with target shapes with high cavities, or if not enough correspondence points between the source and the target shapes have been marked (mesh transfer task only).

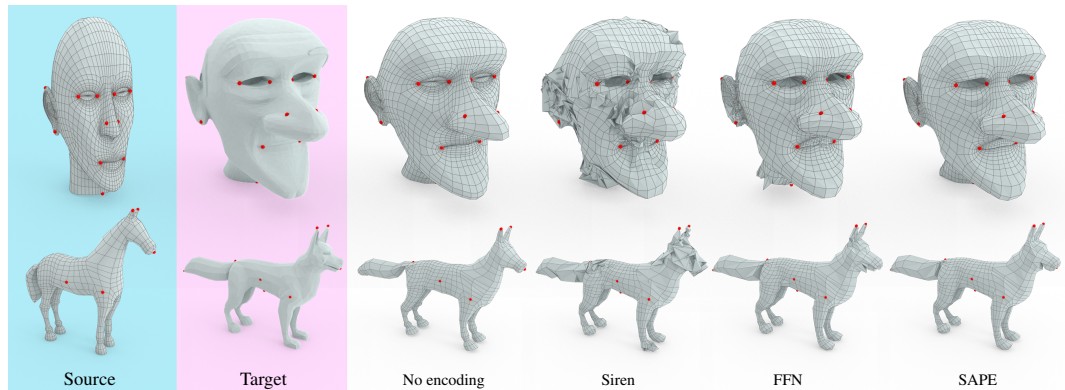

Figure 8: Additional mesh transfer results.

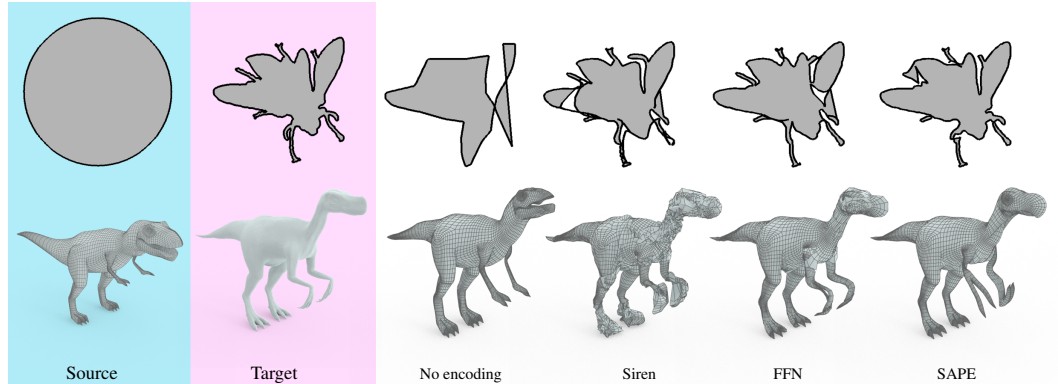

Figure 9: Limitation of the geometric tasks. Without the set of correspondence points for the mesh transfer task, or complex 2D silhouette, our method struggles to fit to the target shape.

# 5 Mesh Transfer Overview

In Section 5.1 we lay out in detail the optimization terms of the Mesh Transfer problem. In Section 5.2 we explain the feedback loss of SAPE for this task.

## 5.1 Optimization Term

In this task, we would like to transfer a 3D source mesh $\mathcal{M}$ to a target shape $\mathcal{T}$, which is represented by a mesh or a point cloud. The MLP receives the vertices of $\mathcal{M}$ and outputs transformed vertices, such that together with the source tessellation, the optimized mesh $\widehat{\mathcal{M}}$ fits the target shape while respecting the structure of the source mesh in terms of its triangulation and local structure.

In addition, we may utilize a set of marked correspondence points $\{v_i, u_i\}_{i=1}^{k}$ between the source mesh and the target shape. The corresponding points enable the estimation of an initial affine transformation from $\mathcal{M}$ to $\mathcal{T}$, followed by a biharmonic deformation [1] in which the corresponding points are set to be the boundary conditions.

The optimization loss for this task is composed of a distance loss and a structural term:

$$\mathcal{L}\left(\widehat{\mathcal{M}} \,|\, \mathcal{T}, \, \mathcal{M}\right) = \mathcal{L}_d\left(\widehat{\mathcal{M}} \,|\, \mathcal{T}\right) + \gamma \mathcal{L}_s\left(\widehat{\mathcal{M}} \,|\, \mathcal{M}\right).$$

The distance loss is given by

$$\mathcal{L}_d\left(\widehat{\mathcal{M}} \,|\, \mathcal{T}\right) = ch\left(\widehat{\mathcal{M}}, \, \mathcal{T}\right) + \sum_{i=1}^{k} \|\widehat{v}_i - u_i\|_2^2, \tag{4}$$

where $ch(\widehat{\mathcal{M}}, \mathcal{T})$ is a symmetric chamfer distance between uniformly sampled points on the optimized mesh and the target shape. In addition, we keep the $k$ correspondence points close by minimizing

the squared distance between them, where $\widehat{v}_i$ and $u_i$ are pairs of corresponding points on $\widehat{\mathcal{M}}$ and $\mathcal{T}$, respectively.

The structural loss measures the discrete conformal energy between the optimized mesh and the source mesh:

$$\mathcal{L}_s\left(\widehat{\mathcal{M}} \,|\, \mathcal{M}\right) = \frac{1}{N} \sum_{i=1}^{N} \left( \sum_{j \in R(i)} \|\hat{\alpha}_j - \alpha_j\|_2^2 \right),$$

where the first summation is over the $N$ vertices of $\mathcal{M}$ and the second iterates on the angles in the 1-ring of $\hat{v}_i$ with respect to the original angles in the source mesh.

To prevent numerical issues caused by *skinny* faces, we utilize the quality measure for a triangular face [3], $Q_f = \dfrac{4\sqrt{3}A_f}{\|e_1\|^2 + \|e_2\|^2 + \|e_3\|^2}$, where $A_f$ is the area of the face and $\|e_i\|$ is the length of its $i$-th edge. When $Q \to 0$, the face approaches to degenerate zero area. To prevent such cases we penalize by $1 - Q_f$ all the faces in $\widehat{\mathcal{M}}$ with quality $Q_f < 0.1$.

## 5.2   Loss Feedback

We adjust SAPE to support the mesh transfer (and also the 2D silhouette) task by using the mesh (polygon for 2D silhouettes) connectivity as the grid which stores the masking parameters $\alpha$.

The progression control is made with regard to the chamfer loss which aggregates the loss of randomly sampled points on the optimized mesh (polygon).

We provide feedback over the loss (equation 7 in the paper) using the barycentric coordinates of the sampled points as the interpolation weights.