# OpenReview forum: "SAPE: Spatially-Adaptive Progressive Encoding for Neural Optimization"
_NeurIPS.cc/2021/Conference — NeurIPS 2021 Poster_

### Official Review · Reviewer_bcJK · 2021-07-13

**Rating:** 7
**Confidence:** 3

**Summary:**

The submission presents a way of improving training of deep implicit networks, that take Cartesian coordinates (e.g., 2D or 3D) as inputs. The method uses additional embeddings of the input coordinates, similarly to Positional Encodings and variations (SIREN, FFN), with 2 differences:
- embeddings corresponding to higher spatial frequencies are only be made available later during the optimization phase, and only if the network does not fit well enough
- they can be made available selectively on different parts of the input space, based on a spatial grid.
This method makes it possible to better represent inputs where the maximal spatial frequency changes a lot between regions (e.g., an image with large smooth regions, and regions of higher detail), and compares favorably to similar methods.

**Limitations And Societal Impact:**

The authors seem to only acknowledge limitations, societal impact, and resource use in the Appendix, rather than the main paper, despite the fact that the page limit was increased specifically to make room for these discussions.

I understand that all the items in the checklist are not supposed to fit in the main paper (especially code and details of experiments), but I think that items starting with "Did you describe" or "Did you discuss" (i.e., 1. (b), (c), 3. (d), 4. (d), (e)) can be answered, at least briefly, in the main paper. If the submission is accepted, I think those discussions should be moved to the main text.

Regarding societal impact, the authors could have mentioned the risk of unfaithful reconstruction of images, as was demonstrated by Xerox incorrectly reproducing characters for instance (https://www.dkriesel.com/en/blog/2013/0802_xerox-workcentres_are_switching_written_numbers_when_scanning), but the statement was overall satisfactory.

**Main Review:**

Originality
---------------
The methods are an improvement on a recent set of techniques in this field, with two additions:
- progressive availability of higher-frequency embeddings, and
- grid-based local availability of these features.
These additions could have been inspired from other fields like image compression: JPEG compression, for instance, also lays a grid on a 2D image, and uses progressively higher-frequency DCT bases.

Related work is correctly cited, explained, and compared against, the novel improvements are clearly described.

Quality
-----------
Experiments are well designed, and support the claim across a variety of tasks: 2D image regression, 2D silhouettes, 3D occupancy, and 3D mesh transfer. Quantitative and qualitative results are shown.

There's some confusion in Table 1 about intervals and significance: It is not clear what the "±" numbers indicate (standard deviation? 95% confidence intervals), what bolding means, and how ties are handled. For instance, on the last column, both "SAPE + RBFG" and "SAPE + FF" have the same exact mean (0.928), but only the latter is bolded.

Clarity
---------
The submission is easy to follow and well written, conveying well the method, and the experiments. With the additional information, I believe someone already familiar with the field of deep implicit networks could reproduce the results.

Significance
-----------------
These results are part of a series of improvements in a recent and quickly-developing field, while maybe not ground-breaking, I believe this method is straightforward enough and widely applicable, and should improve the state of the art.

Suggestions and minor comments
-------------------------

L. 17: "perceptrons"
L. 37 is the first time "implicit neural networks" are mentioned, I think it should be mentioned earlier, maybe in the abstract
L.135, Eq (4) seems to have the condition backward, should the first line say "if i > d"? It would be consistent with l. 141 which says "we set the masks of the *first* d identity encoding functionals as 1"
L. 143: "private case" -> "special case"?
L. 169 : Figure should be numbered
L. 269, Figure 8: It would be better to have consistent axes for the left / right parts, instead of having each encoding be a column left and a row right

After feedback
============

I found the author feedback to this review satisfactory, and I feel like other reviewers' comments were properly addressed.


**Time Spent Reviewing:**

7

---

> ### Author Response · Authors · 2021-08-10
> **Response to reviewer 4**
>
> Thank you for your helpful comments.
>
> **[Notations in Table 1]**
>
> Yes, as you pointed out,, "±"  in our tables refers to the standard deviation of the results and bold means highest score. In the future revision, we will make sure it will be clearly described in the caption of Table 1.
>
> &nbsp;
>
> **[Minor comments]**
>
> Thank you for the small corrections, we will fix them in the next revision.
> You are right about the typo in eq. 4; the inequality symbol should be the other way around and we will fix it.
> In addition, we mention another typo we spotted after submission which we will be fixed in the next revision: $\tau = 2n / T$ should be $\tau = T / 2n$.
>
> &nbsp;
>
> **[Limitations and Societal Impact]**
>
> Thank you for the interesting reference! We can definitely elaborate on the risk of using reconstruction and compression, as implicit representation may have the potential of producing unfaithful results.
> In the revision, we will make sure we move the discussions of limitation, future work and social impact to the main paper.

---

### Official Review · Reviewer_FBfi · 2021-07-14

**Rating:** 8
**Confidence:** 3

**Summary:**

This paper highlights a problem with Position Encoding models that use Fourier features: they're very sensitive to a bandwidth parameter that trades-off between reproduction of low- and high-spatial frequency features. Their solution is an iterative feedback algorithm that reconstructs images in a coarse-to-fine manner. They describe incremental improvements across-the-board, and large and impressive improvements in their 2D silhouette reconstruction task.

**Ethical Concerns:**

No.

**Limitations And Societal Impact:**

Yes.

**Main Review:**

Strengths:

This paper is great. Very clear and well-motivated, with sensible experiments and nice results. Presentation is solid throughout, and the authors also include an extensive supplement. I learned a lot reading this paper, and I think the authors deserve credit for their bulletproof arguments. This should be accepted to NeurIPS, and depending on the other reviewers' thoughts, selected for a spotlight.

I really like Fig 1 as a motivating example. Super clear.


Weaknesses:

The feedback method is very similar to models/objective functions of "predictive coding" in computational neuroscience. It would help to cite those. I think this would make your method even more interesting I think, since it would show that this principle from neuroscience is relevant to the problems you address.

I'm no expert in this field. So my other concerns can be taken with a grain-of-salt. (1) The problem seems like it could be an important one but for most of the experiments the improvements are incremental. There's not much you can do about this, not your fault, just my intuition reading this. (2) I realize the standard in this field is for fitting rather than generalization, but it still seems peculiar to introduce ground-truth data into the inference loop.

**Time Spent Reviewing:**

2

---

> ### Author Response · Authors · 2021-08-10
> **Response to reviewer 3**
>
> Thank you for your feedback and comments.
>
>
> **[Similarity to "predictive coding"]**
>
> Thank you for contributing this inspiring and useful reference!
> We were actually unfamiliar with "predictive coding” in computational neuroscience. After reading about it, we agree that our feedback mechanism bears very nice similarities with this concept, and hope that perhaps future work can mutually benefit both fields. We will definitely add to our paper a couple of sentences discussing this relationship.
>
> &nbsp;
>
> **[Introducing ground truth to inference loop]**
>
> Indeed, neural implicit functions which are heavily discussed in our paper, may seem peculiar at first, as most of us are used to thinking of neural networks in the context of generalization over a large training set.
> In contrast, the recently emerging field of neural implicit functions discusses the usage of neural networks to overfitting complex signals for various applications.
> The original NERF paper by Mildenhall et al. describes one fascinating use case for learning continuous volumetric scene functions.
> In addition, the paper “On the Effectiveness of Weight-Encoded Neural Implicit 3D Shapes” by Davies et al., though not a positional-encoding paper per-se, does a great job at presenting further motivation to neural implicit functions. In particular, we recommend the discussion in their introduction for an insightful read.

---

### Official Review · Reviewer_cagt · 2021-07-16

**Rating:** 7
**Confidence:** 3

**Summary:**

The paper presents SAPE, a progressive (gradually unmasking encoding units from low to high frequency during training) and spatially adaptive (each location's unmasking process can be halted based on its local loss to prevent overfitting) encoding of input signals (to be fitted by an MLP), and demonstrates its superior (visually and numerically) results on various regression tasks and 3D mesh transfer.

**Limitations And Societal Impact:**

Yes.

**Main Review:**

Pros
+ The experimental results are overall impressive (visually and numerically).
+ The writing is clear and easy to follow.

Cons
- The loss-conditioned "linear policy" (Eq 4 & 5), being the key feature of the paper, is not sufficiently studied or justified. It's unclear if the loss-conditioned linear progression is really a good approximation of some "optimal policy" (e.g. learned via RL or meta-learning approaches), which could be critical in this application since the relatively costly training (instead of inference) needs to be run for each new input. In addition, given that the bandwidth requirement for a local region of an input is usually easily estimable (e.g. via DCT/FFT coefficients for 2D images/silhouettes and geometric Laplacian [42] for 3D models), it's unclear how much advantage Eq 5 really provides.
- The application and usefulness of SAPE are not sufficiently discussed or validated in the paper. For example, without direct comparisons against existing 2D/3D compression (and 3D mesh transfer) algorithms, it's unclear if SAPE is really a strong alternative or replacement (or how its 2D/3D regression capabilities can be used in real life). The paper also should have more validation tasks (e.g. NeRF) given that 3D mesh transfer seems to be the only task closer to real-world applications.
- The computational and memory complexities (e.g. wall clock time, number of parameters, etc.) of SAPE should be included, given the amount of algorithmic features (including sparse grid sampling) introduced in the paper.

[42] Spectral Compression of Mesh Geometry, ACM SIGGRAPH, 2000

Post rebuttal
- I'm overall satisfied with the authors' feedback, including the additional details on the storage complexity, the additional results on tiny-Nerf, and the additional discussion of the motivation and applicability of SAPE referencing Davies et al. I've upgraded my score to 7.

**Time Spent Reviewing:**

2

---

> ### Author Response · Authors · 2021-08-10
> **Response to reviewer 2**
>
> We thank you for the constructive comments.
>
> **[Masking policy]**
>
> You're correct to point out that in our setting, each new input requires re-training, and it is therefore critical to keep the optimization efficient. Yes, the unmasking policy is linear and simple, but at the same time it is also quite efficient as we show throughout our experiments.
> Our goal is to keep our method useful for a variety of optimization tasks. In most cases, we found that using a linear policy already gives satisfactory results, and doesn't justify employing resource intensive tools such as RL and meta learning to search for the optimal policy.
> That being said, these could definitely be great research directions for future works.
> In addition, as also requested by Reviewer 1- SbWq, we will add an ablation experiment to support the justification of spatial progression compared to a global progression policy (see the results of this ablation in the answer to R1).
>
> &nbsp;
>
> **[Bandwidth could be estimated]**
>
> We acknowledge that the spatial bandwidth can be estimated for each task separately. Yet, this adds a burden in the usage of our method. Therefore, in our work, we aimed for a general framework that can be easily injected into different tasks.
>
> &nbsp;
>
> **[Discussion of applications]**
>
> SAPE is a method which attempts to tap into the full potential of neural networks using positional encodings, so naturally use cases involving neural implicit functions are immediate benefactors in terms of output quality.
>
> To demonstrate the strength of SAPE in terms of output integrity, we suggest the following walkthrough:
> Begin by observing the chirp signal in Figure 2.
> You can see that non-adaptive positional encodings tend to learn an implicit function which overshoots around the smooth areas of the signal, which is dominated by lower frequencies.
> The 1d signal example is a good starting point to gain intuition, because the same phenomenon is evident in other media, as we’ll soon illustrate, but is less perceived at first glance.
>
> Next, we advise to review the results in Table 1 & 2 in the context of visual results, as differences in some quantitative metrics like PSNR and IoU, may not perfectly correlate with differences in visual quality.
> For example: compare the 3d occupancy IoU in Table 1 with Figures 6,7 in the appendix.
> These are high-resolution models, and therefore zooming in is highly recommended.
> Note the Turbosquid samples of Figure 7, where SIREN fails to capture the side of the house, or the staircase of the pagoda. Here, FFN requires heavy tuning to avoid artifacts (for complex shapes, this process can get tedious).
> You can also notice, i.e, that for the statue sample in Figure 6, both SIREN and FFN exhibit “ripples” at the smooth areas of the shape (i.e: at the cheek right below the eyes, or around the shoulders). This phenomenon is similar in nature to the behaviour we observe in Figure 2.
>
> In contrast, the progressive and adaptive nature of SAPE minimizes the effect of such artifacts, and our method is able to provide detail preserving outputs of high quality.
> We also remark that we did study the possibility of adjusting our method to support the NeRF task, and observed an improvement over the baselines methods. We detailed the setting of our experiment and our preliminary results in our reply to R1.
> Finally, we mention that SAPE unlocks the usage of learned neural nets in geometric tasks with indirect supervision (e.g: 2d silhouettes and 3d mesh transfer), where existing positional encoding methods currently don’t provide adequate quality (Figures 8 in the paper and the appendix, Tables 1 & 2). At the time of writing these words, these research areas are dominated by traditional methods (i.e: surface mapping via parameterization). As the scope of our paper is limited to the study of neural positional encoding, we chose to compare with other neural methods which are closer in spirit to our work.
> We think that further study of how such geometric tasks may benefit from neural methods, or even combine with traditional methods, is a very interesting research direction.
>
> Generally speaking, SAPE is applicable where other learned implicit functions are applicable. Since the size of the network is orthogonal to our suggested policy, we do not claim to provide improved compression over other neural implicit methods.
> Within the domain of neural implicit functions itself, SAPE inherits the benefits of other learned implicit functions over traditional formats like meshes.
> In that regard, the work of Davies et al., though not a positional-encoding paper per-se, does a great job at presenting further motivation to neural implicit functions (see: “On the Effectiveness of Weight-Encoded Neural Implicit 3D Shapes”). We recommend the discussion in their introduction for an insightful read.
>
> To provide stronger motivation, we will expand on the background and applicability of neural implicit functions in our introduction.
> In addition, In our revision, we will make sure to include visual insets to better guide readers through these results similar to the walkthrough we provided above.
>
> &nbsp;
>
> **[Complexity]**
>
> We report the run times and exact training parameters for each evaluation in the appendix under Section 5. To increase the visibility of implementation details, we will mention the primary details as well as the storage complexity in the main paper.
>
> Spatial adaptiveness requires keeping track of an additional parameter per spatial location of the signal.
> However, the grid of masks was introduced to mitigate this added memory complexity (rather than being the cause of it). Practitioners may configure the resolution of this grid to control the tradeoff of spatial-adaptiveness vs. added complexity.  At the extreme case of resolution of 1, our method has no added complexity, and is progressive, but not spatial adaptive. Figure 7 in the paper discusses this tradeoff.

---

### Official Review · Reviewer_SbWq · 2021-07-19

**Rating:** 5
**Confidence:** 4

**Summary:**

This paper proposes an improvement of implicit function approximation for image and mesh generation. The main contribution is to use a progressive low resolution to high resolution optimization as well as spatially adapt this `masking`. Performance is shown to be better than FFN by a significant margin.

**Limitations And Societal Impact:**

yes.

**Main Review:**

This paper proposes an improvement of implicit function approximation for image and mesh generation. The main contribution is to use a progressive low resolution to high resolution optimization as well as spatially adapt this `masking`. Performance is shown to be better than FFN by a significant margin.

The positives of this paper is that it achieves superior results in terms of PSNR, IOU, and qualitatively over different datasets. The technique makes sense and is relatively straightforward. The paper is also clearly written and the figures provide intuition as to why this method works.

There are a couple of negatives to the paper, which if addressed will help the paper improve.
- The curriculum of the mask as training progresses is mostly ad hoc and not set according to any objective. Could they be optimized in some way, similar to meta-learning objectives perhaps.
- Experimental results do not have enough ablative studies. Specifically, the decisions made for Eq. 4 and spatially adapting the \alpha vector is not justified empirically. Line 273 shows a qualitative and anecdotal image comparing the approaches, quantitative results will make a stronger point. It would be good to quantify the design choices of SAPE.
- Experimental training parameters and settings are not mentioned in the main paper. (it would be nice to have a brief discussion in the main paper, e.g. how long does it take to train, how many parameters are there.)
- How does the number of bits required for the network compare to the number of bits for the image itself? In other works, how inefficient/efficient is it to store images via implicit function as compared to just storing a jpeq image.

Questions:
- How would this approach work with NERF family of methods.
- Line 143: private case -> specific case ?

Overall, this is a nice paper with good empirical results. However the lack of ablative experiments and the lack of network details is a negative for the current version of the paper.







**Time Spent Reviewing:**

2

---

> ### Author Response · Authors · 2021-08-10
> **Response to reviewer 1**
>
> Thank you for reviewing our paper thoroughly and providing insightful comments.
>
> **[Mask progression / meta-learning]**
>
> In this work, we showed that a “simple” masking policy is already efficient enough to improve various tasks, and therefore focused on the ablation studies of the encoding frequency band and grid size of the spatial progressive mask..
> However, we also conducted additional ablations on the tasks of Table 1 to quantitatively validate the contribution of the spatial progression. Specifically, we ran the tests using global progression on the Fourier Features encodings, similar  to [16], [26].
> For the image regression task, the global progression led to  a drop of ~1dB (27.1, 30.2 PSNR on natural, text images respectively).
> For the 3D occupancy task, we observed a large drop on the more complex datasets (0.98, 0.943 IoU on Thinki10k, Turbosquid respectively).
> For  the 2D silhouettes task, a FF network with global progression achieves  0.86 IoU: which is a drop of 0.06.
> We will include these additional ablations in the paper.
> Applying meta-learning to optimize the masking progression policy over several objects (or tasks) is an interesting direction for future work; we will add such a comment in the revised version under discussion of future work.
>
> [16] Barf: Bundle-adjusting neural radiance fields.
> [26] Deformable neural radiance fields.
>
> &nbsp;
>
> **[Experimental settings]**
>
> We included the implementation details in the appendix due to space constraints, but we can definitely move the important settings and implementation details (i.e. training time and number of parameters) to the main paper:
> Note that as we described in the appendix, in this work we experiment with relatively small networks (up to 4 fully connected layers) which maintain short training times.
> For reproducibility, we have included an initial draft of the code with this submission. We also plan to publish the full codebase.
>
>
> &nbsp;
>
> **[Storage]**
>
> The storage capacity in our implementation of a SAPE implicit image function is 1.82 MB compared to 1.01 MB of FFN / SIREN network. Spatial adaptiveness requires keeping track of an additional parameter per spatial location of the signal.
> Therefore, we introduced the grid of masks to mitigate this added memory complexity. Practitioners may configure the resolution of this grid to control the tradeoff of spatial-adaptiveness vs. added complexity.  At the extreme case of resolution of 1, our method has no added complexity, and is progressive, but not spatial adaptive. Figure 7 in the paper discusses this tradeoff.
>
> In addition, we note that implicit representations are sometimes employed for other means and not just compression. For example in the NERF line of works, implicit representations reconstruct a full scene from a set of multi-view images. In our work, we used it for different tasks, such as reconstruction from undersampled images and shapes, as well as geometric tasks like mesh transfer.
>
> &nbsp;
>
> **[NeRF group of works]**
>
> Extending NeRF wasn’t the focus of our work, but rather the positional encoding. Nevertheless, we researched how our method may be adapted to support the setting of “NeRF-like” optimizations, and Following the reviewer’s request, we tested this approach on the same “tiny-NeRF” test and settings as in [35].
> Our adaptation to Nerf-like is based on the 3D Occupancy task where we keep track of a 3D masking grid. During the optimization, the loss is received for a 2D pixel projection based on 3D sampled points along a ray. In the feedback stage (Fig.3), we propagated this loss to the sampled points.
>
> For the tiny-Nerf experiment, we got improved results:
>
> FF - $25.62 \pm 0.87$, Progressive FF - $25.58 \pm 0.9$ and **SAPE + FF: $25.94 \pm 0.88$**,
>
> where the measurements are the mean PSNR of the test $\pm$ the standard deviation.
>
> [35] Fourier features let networks learn high frequency functions in low dimensional domains.

---

### Decision · Program_Chairs · 2021-09-28

**Decision:**

Accept (Poster)

**Comment:**

This paper presents a coarse-to-fine scheme for learning coordinate-based neural networks that is spatially adaptive to the content of the image. While similar coarse-to-fine schemes have been previously presented for NeRF, the spatial adaptivity is shown to improve results across a variety of applications, and reviewers appreciated the clarity of the writing and presentation. The strong empirical results along with quality of presentation lead me to recommend accepting this paper.

**Consistency Experiment:**

NeurIPS has a long history of experimentation. In 2014, NeurIPS ran an experiment in which 10% of submissions were reviewed by two independent committees to quantify the randomness in the review process. This year, we repeated a variant of this experiment to see how the quality of the review process has changed over time.  This paper was part of the experiment and was therefore assigned to two committees (consisting of reviewers, an Area Chair, and a Senior Area Chair) that reached independent decisions.  If both committees made the same recommendation, this recommendation was followed. If a single committee recommended acceptance, the paper was accepted (with the exception of a few cases in which the other committee identified what we considered a fatal flaw, e.g., an error in a key result).

Both committees reached the same decision: **Accept (Poster)**

The other committee assigned to the paper recommended **Accept (Poster)**.  You can find the other set of reviews, along with any follow up discussion with the authors here:
https://openreview.net/forum?id=_wPmKqEMxss